Adapting multilingual vision language transformers for low-resource Urdu optical character recognition (OCR)

Cheema Musa Dildar Ahmed 1
Shaiq Mohammad Daniyal 1
Mirza Farhaan 2
Kamal Ali 1
Naeem M. Asif 1 asif.naeem@nu.edu.pk
1 Department of Artificial Intelligence and Data Science, National University of Computer and Emerging Sciences , Islamabad , Pakistan
2 School of Computer, Engineering and Mathematical Sciences, Auckland University of Technology , Auckland , New Zealand
Raza Khalid
Electronic publication date: 2024 Apr 29
Publication date: 2024
Volume: 10
Electronic Location ID: e1964
Received 2023 Oct 17; Accepted 2024 Mar 6
Copyright: © 2024 Cheema et al.
Copyright year: 2024
Copyright holder: Cheema et al.
License: This is an open access article distributed under the terms of the Creative Commons Attribution License, which permits unrestricted use, distribution, reproduction and adaptation in any medium and for any purpose provided that it is properly attributed. For attribution, the original author(s), title, publication source (PeerJ Computer Science) and either DOI or URL of the article must be cited.
License URL: https://creativecommons.org/licenses/by/4.0/

Keywords: Document analysis, OCR, Urdu OCR, Multilingual, Transformer based models, Performance evaluation

Funding: The authors received no funding for this work.

==============================
In the realm of digitizing written content, the challenges posed by low-resource languages are noteworthy. These languages, often lacking in comprehensive linguistic resources, require specialized attention to develop robust systems for accurate optical character recognition (OCR). This article addresses the significance of focusing on such languages and introduces ViLanOCR, an innovative bilingual OCR system tailored for Urdu and English. Unlike existing systems, which struggle with the intricacies of low-resource languages, ViLanOCR leverages advanced multilingual transformer-based language models to achieve superior performances. The proposed approach is evaluated using the character error rate (CER) metric and achieves state-of-the-art results on the Urdu UHWR dataset, with a CER of 1.1%. The experimental results demonstrate the effectiveness of the proposed approach, surpassing state of the-art baselines in Urdu handwriting digitization.

Introduction

Optical character recognition (OCR) (Rao et al., 2016) has been a widely researched area since the advent of digital technology. The goal of OCR is to convert scanned images of text into machine-readable text, enabling the digitization and analysis of large amounts of unstructured handwritten or printed text. OCR has its roots in the 1960s and 1970s, where researchers attempted to recognize characters using rule-based approaches and hand-engineered features. With the advent of deep learning, OCR has seen significant advancements in recent years. The first wave of deep learning-based OCR models used convolutional neural networks (CNNs) (Chauhan, Ghanshala & Joshi, 2018) with long short-term memory (LSTM) networks (Graves, 2012). This architecture was improved with the use of Region-Based CNNs (RCNNs) (Bao et al., 2016), which used region proposals for improved feature extraction. However, these models have a major limitation in the form of needing a separate language model at the end, as they are trained on a character by character basis, not on a word level. The disadvantages of a separate language model are the added error of the language model itself, and the added complexity and computational costs. This led to the development of Transformer-based models for OCR, such as Transformer-based OCR (TrOCR) (Li et al., 2021). This replaces the CNN layers with a pre-trained Vision Transformer and uses a pre-trained language model for text generation. This has shown improved performance in terms of accuracy and speed compared to previous models.

Despite the advancements in OCR, the field still faces several challenges, particularly in the case of multilingual documents. The aforementioned disadvantages of previous approaches, plus a lack of parallelizability and scalability of previous models makes it difficult to handle large multilingual datasets. Moreover, the accuracy of TrOCR models depends on the pre-training of the language model, which may not always reflect the language and script of the target document. This problem is particularly pronounced in the case of non-Latin scripts such as Arabic or Urdu, where traditional OCR models perform poorly.

This research article focuses on the digitization of bilingual documents, with a specific emphasis on low-resource languages. It aims to investigate the challenges and opportunities of digitizing low-resource languages, analyzing existing approaches, tools, and technologies used in the process. Then we propose a solution to the problem of digitization of handwritten bilingual medical prescriptions, which contains both English and Urdu (a low resource language) text. We address the limitations of previous OCR models by proposing a model that is specifically tailored for bilingual text extraction. To demonstrate the viability of our approach, we use a synthetically generated handwritten bilingual medical prescription data set as a case study, coupled with other publicly available Urdu and English datasets. The medical domain is a particularly interesting area to study as it requires accurate and efficient digitization of information contained in handwritten prescriptions. Our contributions in this article can be summarized as follows: ViLanOCR: We introduce a novel modular approach for digitizing bilingual (English and Urdu) handwritten prescriptions. The proposed approach is specifically tailored for an unconstrained environment, reflecting its practical relevance in real-world scenarios. Moreover, by incorporating bilingual capabilities, our approach caters to the unique requirements of documents written in both English and Urdu, making it adaptable to a broader range of other use cases and languages.

Dataset: We develop a comprehensive dataset for Urdu language, by combining our own synthetically generated medical prescriptions dataset with augmented pre-existing publicly available datasets.

Performance evaluation: We evaluate our approach with stat-of-the-art models. In our experiments VilanOCR achieves character error rate (CER) of 1.1% on the UHWR dataset, serving as a strong benchmark for evaluation.

The rest of the article is structured as follows. “Related Work” presents a detailed literature review on existing OCR methodologies, focusing on bilingual/multilingual approaches. “Proposed Approach” presents our proposed approach, ViLanOCR. “Experiments” presents dataset and model & methods. “Results and Discussion” presents results and discussion. “Extended Study” presents extended study of our proposed model. Finally “Conclusion and Future Directions” concludes the article and provides some future directions.

Related work

There are several existing approaches for digitization of documents in English language, which could be used as a foundation for developing approaches for digitizing bilingual documents. However, digitization of bilingual documents containing languages with altogether different scripts such as English and Urdu is a challenging task. One major challenge is the difference in scripts, which requires handling and recognition of diverse character sets. Another challenge is the lack of parallel data for low-resource languages. This makes it difficult to train models for digitization of documents containing these languages.

Tesseract (Rice, Jenkins & Nartker, 1995) is an open-source OCR engine supported and maintained by Google, using a combination of robust algorithms and techniques. The engine utilizes a multi-stage process that includes image preprocessing, layout analysis, text line extraction, character segmentation, and optical character classification. These techniques encompass methods such as thresholding, connected component analysis, adaptive recognition models, and statistical language modeling. Tesseract underwent a significant transformation with the release of version 4, where it introduced a new LSTM engine for improved performance. LSTM is a type of recurrent neural network that excels in handling sequential data, making it well-suited for text recognition tasks. The integration of the LSTM engine empowered Tesseract to handle complex text layouts, handle recognition errors more effectively, and achieve higher accuracy rates compared to its earlier iterations (Smith, 2016). In this, the authors also attemtped to adapt the now LSTM powered Tesseract OCR for multilingual use cases. However, the character error rate (CER) and word error rate (WER) for non-Latin languages, particularly Indic languages, was less than optimal (e.g. 25% WER for Hindi, with the LSTM + Dictionary approach).

EasyOCR boasts a wide language support, encompassing over 80 languages (JaidedAI, 2023), making it exceptionally versatile for multilingual OCR tasks. It incorporates several preprocessing steps, such as grayscale conversion and other relevant techniques, within its library to enhance the quality of input images before performing text extraction. Moreover, it uses the Character Region Awareness for Text Detection (CRAFT) (Baek et al., 2019) algorithm for efficient text detection. The recognition model employed by EasyOCR employs the convolutional recurrent neural network (CRNN) architecture. The CRNN utilizes a combination of LSTM and connectionist temporal classification (CTC) to perform sequence labeling. LSTM, a type of recurrent neural network, handles the sequencing aspect, while CTC is responsible for labeling unsegmented sequence data. This combination allows EasyOCR to accurately recognize and extract text from a variety of input images. However, even though the authors claim support for Urdu, EasyOCR does not provide acceptable results for our use case, unfortunately.

PaddlePaddleOCR (PPOCR) (Du et al., 2020), an open-source OCR framework, has emerged as a powerful solution due to its robust performance, versatility, and extensive language support. It incorporates state-of-the-art deep learning models, such as Faster R-CNN, Mask R-CNN, and CRNN, to perform text detection, recognition, and layout analysis tasks. PPOCR’s modular design, flexible configuration options, and ease of integration have made it popular among researchers and developers. PPOCR exhibits significant potential for multilingual OCR applications. Its language support spans a wide range of scripts and writing systems, including Latin, Chinese, Cyrillic, and Arabic. PPOCR’s architecture and training methodologies enable effective adaptation to different languages, making it well-suited for multilingual OCR tasks.

Li et al. (2021) propose an innovative architecture for OCR, TrOCR, which utilizes pre-trained transformers. In this approach, the authors eliminate the use of CNNs traditionally employed for image feature extraction in OCR tasks. Instead, they leverage pre-trained vision transformers that draw inspiration from self-attention mechanisms commonly utilized in natural language processing (NLP) tasks. Unlike CNNs, which operate on the entire input image matrix, vision transformers break down the image into patches. These patches are treated as tokens and passed as input to the vision transformer model. In subsequent layers, the model learns attention weights for each token, generating embeddings that represent the input image in a latent space. The TrOCR model employs a pre-trained language model, which obtains embeddings from the vision transformer model and generates output tokens for the input image. Training occurs in an autoregressive manner, wherein both the vision transformer and the language model are fine-tuned based on the OCR input and output features. Previous approaches to OCR involved a CNN-to-LSTM architecture for text extraction. However, this approach had limitations. It relied on an LSTM network, which is not easily parallelizable for scalability and does not effectively handle long-term dependencies. In contrast, TrOCR incorporates a RoBERTa-based (Liu et al., 2019) language model, which can be parallelized. Additionally, transformer-based language models have demonstrated state-of-the-art performance in text generation tasks, and utilizing them for OCR enhances the overall system performance. Previous OCR models required a separate language model for text correction, but TrOCR’s architecture integrates the language model, eliminating the need for an additional component. TrOCR achieves state-of-the-art results in English OCR tasks and is considered one of the best OCR systems for handwritten text. However, it is important to acknowledge a weakness in this work, which lies in the computational overhead of transformer models employed in TrOCR. The use of two large transformer models, one for vision and one for text generation, results in increased inference time compared to previous approaches. Moreover, TrOCR’s accuracy is dependent on the pretraining quality of the language model it utilizes.

In their study, Tabassum et al. (2022) highlight the time constraints faced by doctors in developing countries, which hinder their ability to write digital prescriptions. They point out that the World Health Organization (WHO) recommends an ideal doctor-to-population ratio of 1:1,000, while in developing nations such as Pakistan, Bangladesh, and India, the ratio stands significantly lower at 0.304:1,000 (Kumar & Pal, 2018). Consequently, the majority of doctors in these countries still rely on handwritten prescriptions due to the high cost of digital alternatives and a lack of IT literacy. However, digital prescriptions have gained global popularity for providing easier access to medical records. To address this issue, Tabassum et al. (2022) propose an end-to-end recognition system that employs a Bidirectional Long Short-Term Memory (BiLSTM) approach to digitize multilingual handwritten prescriptions. They introduce a dataset named the “Handwritten Medical Term Corpus,” consisting of 17,431 samples comprising 480 medical terms. Various augmentation methods and image preprocessing techniques are utilized to enhance the accuracy of the system. By combining the BiLSTM approach with data augmentation techniques such as rotating, shifting, and stretching, the authors achieve state-of-the-art results. Their approach achieves an average accuracy of 93%, representing a substantial improvement over previous methods. However, it is important to note a significant limitation of the proposed approach. It requires doctors to use a smartpen for writing prescriptions, which may not be practical or feasible for many clinics and hospitals. Moreover, the proposed system may have limitations in recognizing pre-existing prescriptions, which could restrict its applicability in certain settings.

In their research, Anjum & Khan (2020) delve into the realm of offline Urdu handwritten text extraction, proposing a novel dataset and employing a specialized Convolutional Recurrent Neural Network (CRNN) architecture. Their dataset, a significant contribution in itself, comprises 7,300 lines of text penned by 100 distinct writers. The core of their approach is a CRNN that integrates a Densenet architecture for initial image feature extraction, chosen for its effectiveness in retaining information across layers, akin to a residual network. This is followed by a Gated Recurrent Unit (GRU) for text generation, augmented with an attention mechanism. The attention is particularly crucial given the unique contextual dependencies of Urdu characters, where their shape varies based on location within a word. This methodology yielded a character-level accuracy of 77.5% and a word-level accuracy of 43.5% on their dataset, underscoring its efficacy albeit with room for improvement, particularly at the word level.

Similarly, in their article, ul Sehr Zia et al. (2022) address the challenge of digitizing handwritten Urdu text from images. They introduced the Urdu Handwriting dataset (UHWR), a substantial dataset encompassing over 70% of Urdu script variations. The complexity of Urdu’s cursive nature and its extensive ligature repertoire underscores the significance of this dataset. Their methodological approach entailed an image-to-sequence framework, combining a modified CNN with a bidirectional long short-term memory (BiLSTM) network for text translation. A notable innovation in their CNN design was the replacement of traditional max pooling with concatenation in intermediate layers, a strategy aimed at reducing information loss. The efficiency of their approach was quantified using the CER metric, achieving a noteworthy 7.35% CER on the UHWR dataset. While their model demonstrated potential for cross-lingual applicability, its focus was confined to individual text lines, not extending to the digitization of full documents. This limitation, along with the necessity of high-resolution images and a separate language model for error correction, could pose challenges in broader, resource-constrained applications.

In their recent publication, Fateh, Fateh & Abolghasemi (2021) introduce an innovative approach to digitize multilingual handwritten numbers. The authors emphasize the significant interest in recognizing handwritten numerals within the computer vision and image processing communities. They argue that multilingual numeral recognition systems hold great importance in today’s world, with increasing international correspondence and transactions, especially in multilingual countries where multiple languages coexist. However, the development of such systems poses challenges due to the need to handle diverse writing styles and languages with distinct scripts. To tackle these challenges, the authors propose a language-independent model composed of two distinct components: language recognition (LR) and digit recognition (DR). The LR model is responsible for classifying the input sample based on its language, employing a CNN architecture for language localization. Once the language is identified, the LR model forwards the sample to the appropriate DR model. The DR model, also utilizing a CNN architecture similar to the LR model, focuses on digitizing the handwritten text. Transfer learning techniques are employed to improve image quality and recognition performance. To assess the effectiveness of their proposed approach, the authors evaluate it on a diverse set of six languages: English, Arabic, Chinese, Persian, Kannada, and Urdu. The experimental results demonstrate state-of-the-art performance, with an average accuracy of up to 99.8%. One notable strength of the approach is its scalability, as it allows for the inclusion of additional languages without requiring substantial restructuring of the entire system. However, it is important to note that the proposed approach solely caters to digitizing numbers and cannot be applied to handwritten text. Furthermore, the LR model may encounter challenges in differentiating between languages that exhibit similar number styles (for example, the number ‘3’ is written in exactly the same way in Arabic, Persian and Urdu).

Proposed approach

In this section, we present our innovative approach, ViLanOCR, for the extraction of handwritten bilingual text from documents. It adapts a multilingual vision language transformer model to effectively extract both English and Urdu text from images. This section delves into the architecture and mechanics of ViLanOCR, highlighting its key components and methodologies. The pictorial representation of the architecture is shown in Fig. 1.

Figure 1 ViLanOCR architecture.

To validate the efficacy of ViLanOCR, we focus on the domain of handwritten medical prescriptions. These documents encompass a plethora of information, including diagnoses, patient names, ages, medicine names, and dosages, often presented in both English and Urdu. To evaluate the robustness and accuracy of our approach, we use the CER metric. The CER is a measure of the accuracy of text recognition, calculated as the percentage of incorrect characters (insertions, deletions, substitutions) in the OCR output compared to the ground truth. The lower the CER, the more accurately the model transcribes handwritten text. The task of developing a multilingual OCR system, especially for languages with limited resources like Urdu, poses considerable challenges. The scarcity of publicly available handwritten Urdu data restricts the application of large-scale transformer-based models, which have showcased exceptional performance on well-resourced languages.

To address the scarcity of data, we combined self-collected data from 30 individuals. In addition to this, we integrated publicly accessible datasets such as NUST-UHWR (ul Sehr Zia et al., 2022), PUCIT-OHUL (Anjum & Khan, 2020, 2023) and the printed Urdu vocabulary dataset. The combination of diverse datasets contributes to a more robust and versatile model. However, as data quantity alone does not suffice for high accuracy, we employed a suite of augmentation techniques to enrich the dataset. These techniques encompass pixelation, gaussian noise, salt and pepper noise, quantization noise, and shearing, effectively mimicking the variations encountered in real-world handwritten text.

Central to our approach is the fusion of two potent models: the Swin encoder (Liu et al., 2021) and the mBART-50 decoder (Tang et al., 2020). The Swin encoder forms the foundation of our approach, serving as the backbone feature extractor of our approach. It adeptly captures intricate details, ensuring robust feature extraction. A salient feature of the Swin encoder is the incorporation of a sliding window mechanism. This mechanism enables the model to effectively scan the input image in a sequential manner, capturing context while maintaining computational efficiency. Unlike vanilla vision transformers, a Swin transformer uses a shift window attention, which reduces the attention mechanism to work in a small proximity which makes it useful for dense tasks such as object detection and segmentation. Our problem is also a dense task, which requires a pixel level understanding as in handwriting text a minor change can have an effect on the text.

The Swin transformer addresses these variations in handwriting through its shift window partitioning, which localizes attention within smaller regions, thus catering to the little changes in handwriting. It does so by dividing the input image into non-overlapping patches, each treated as a token. The features of these patches are derived from their raw pixel values. For instance, with a patch size of 4 × 4, the feature dimension of each patch is 48 (4 × 4 × 3). These features undergo a linear embedding to project them into a latent space. Swin Transformer blocks (which are modified versions of traditional Transformer blocks) are then applied to these tokens. These blocks maintain the number of tokens while reducing their spatial dimensions through patch merging layers. Each patch merging layer combines features of neighboring patches and applies a linear layer to these combined features, thereby reducing the number of tokens and doubling the feature dimension. This process creates a hierarchical representation of the input image, akin to the layers in conventional CNN networks like VGG or ResNet. As aforementioned, a distinctive feature of Swin Transformers is their use of shifted window partitioning. In simple terms, while one block of the Transformer uses a regular window partitioning, the next block shifts the windows slightly. This shifting introduces connections between neighboring windows, enhancing the model’s ability to capture relationships across different parts of the image. This approach is vital for tasks requiring detailed and contextual understanding of the image, like handwritten text in our case. Additionally, Swin Transformers incorporate a relative position bias in their self-attention calculations. This bias allows the model to better understand the relative positioning of different patches within a window, further enriching its capability to process complex images, especially complex spatial relationships like Urdu handwritten text.

Another motivation for employing the use of the Swin encoder is its efficiency as compared to traditional Multi-Head Self-Attention (MSA). In a traditional global MSA, the complexity is quadratic to the number of patches in the image. In contrast, the Swin Transformer employs window-based MSA, where self-attention is computed within each local window of size M × M. This results in a linear complexity, making it more scalable and efficient, particularly for high-resolution images or dense tasks like handwriting recognition.

The mBART-50 decoder, on the other hand, possess a remarkable capacity to handle diverse scripts and languages. A defining characteristic of the mBART-50 decoder is the employment of multihead attention, a mechanism that endows the model with the ability to focus on different parts of the input image simultaneously. This is particularly crucial for bilingual text extraction, as it enables the model to concurrently process both English and Urdu components, even when they are spatially intertwined. The mBART-50 decoder possesses a distinctive denoising capability, a feature that plays a pivotal role in enhancing the accuracy and resilience of our ViLanOCR architecture. mBART-50 underwent pretraining as a denoising autoencoder. This pretraining regimen involves the assimilation of text from a concatenation of datasets encompassing N languages, where each dataset is a compendium of monolingual documents. The pretraining process is characterized by the implementation of two distinct noise applications: random span masking and order permutation. These methodologies are instrumental in ingraining the model with a robust and nuanced understanding of each language’s idiosyncrasies, a capability that is paramount for the effective decoding of diverse scripts and languages, an essential requirement for the ViLanOCR. Denoising, in this context, involves a deliberate process of introducing random masking to portions of the input image containing text. This masking effectively obscures certain sections, simulating the presence of noise or distortion that is commonly encountered in real-world handwritten text. The task of the mBART-50 decoder is to then reconstruct or “decode” the masked sections, generating the missing text and restoring the original content. The working of Vision Language based OCR (ViLanOCR) is further elaborated in “Model Experimentation”.

Experiments

In this section, we present the experimental results of ViLanOCR. The efficacy of our model is rigorously examined across an array of diverse datasets, encompassing both publicly accessible repositories and our self-curated synthetic medical prescriptions dataset. This comprehensive evaluation encompasses the following noteworthy datasets: IAM Handwriting Dataset (Marti & Bunke, 2002)

Wiki Dataset (Davis et al., 2020)

Synthetically Generated Printed Dataset for Urdu (generated by ourselves)

NUST-UHWR (ul Sehr Zia et al., 2022)

PUCIT-OHUL (Anjum & Khan, 2020)

NUST-UPTI (ul Sehr Zia et al., 2022)

Custom Medical Prescriptions Dataset (generated by ourselves)

These datasets are meticulously chosen and synthesized to provide a comprehensive assessment of our proposed models in real-world scenarios. Notably, the incorporation of our bespoke Custom Medical Prescriptions Dataset not only enriches the diversity of our evaluation but also accentuates our capacity to address bilingual contexts adeptly. This unique dataset augments the versatility of our approach, enabling a rigorous assessment of our model’s effectiveness in bilingual transcription challenges. For a nuanced comprehension of our training and evaluation protocols, we direct our focus to the ensuing sections.

Datasets

Custom medical dataset

To address the scarcity of publicly available medical prescription datasets, especially in the bilingual domain, we developed a custom dataset of synthetic medical prescriptions for both English and Urdu. This dataset contains a variety of medical terms, drug names, and dosage information, making it suitable for training models for medical prescription recognition. Around 1,000 medical prescriptions were synthetically generated by volunteer interns, resulting in 16,992 and 748 samples for English and Urdu handwriting respectively. These were then individually transcribed and labelled, for various different formats (such as B-tagging and I-tagging for BERT). The samples were then augmented, to further enrich the existing dataset, bringing the total number of samples to 50,000.

English datasets

We used a combination of publicly available handwritten and printed text datasets to create a comprehensive dataset for English text recognition. This included the HW_wiki dataset (Davis et al., 2020) which contains over 2 million handwritten words from various sources, such as Wikipedia articles and academic articles. To further improve the diversity of the English dataset, we also used the IAM dataset (Marti & Bunke, 2002), which contains handwritten English text samples from various writers and genres, such as letters, diaries, and forms. Additionally, we incorporated the SROIE dataset (Huang et al., 2019), which contains a mixture of handwritten and printed text samples of receipts, invoices, and other business documents. This allowed us to train our models on more practical and real-world data for text recognition. For printed text, we used the TRDG synthetic dataset (Belval, 2019), which contains over 2 million synthetic printed text samples.

Urdu datasets

Similarly, for the Urdu dataset, we carefully selected both handwritten and printed publicly available datasets. The NUST-UHWR (ul Sehr Zia et al., 2022) and PUCIT-OHUL (Anjum & Khan, 2020, 2023) datasets were used for handwritten text, while the UPTI dataset (which contains 0.5 million samples) and UPTI v2 dataset (which contains 1 million samples) were used for printed text. We made sure to select datasets that are diverse and representative of the Urdu language, and we took care to ensure that the data was appropriately labeled and formatted for our research purposes.

Augmentation

We utilized extensive image augmentation techniques to further enrich and increase our dataset, bringing its size from originally 20 to 200 GB.

The image augmentation techniques we employed serve to enhance and modify the visual characteristics of input images. They employ various techniques to achieve different effects. To introduce blur effects, we utilized filters such as a general blur filter, Gaussian blur, and box blur. These filters apply smoothing operations to the images, resulting in a softer appearance. By adjusting the blur radius, different levels of blurriness can be achieved, allowing flexibility in the desired outcome. We used contrast adjustment to modify the contrast of the images. By utilizing an image enhancement technique, it enhances or reduces the difference between light and dark areas. The degree of contrast adjustment is controlled by a parameter that randomly selects a contrast value within a specified range. Another transformation technique we used is pixelation, where the images are resized to a smaller size and then scaled back up to the original dimensions using a nearest-neighbor interpolation technique. This process reduces the level of detail in the images, resulting in a pixelated effect. Rotation and perspective transformations were also utilized. The rotation function randomly rotates the images within a specified range of angles. This transformation introduces variations in orientation. The perspective transformation function distorts the images by skewing them horizontally, simulating a perspective change. Noise can be added to the images using the noise function. We used different noise types, including Gaussian, salt-and-pepper, Poisson, and speckle. Each noise type introduces different patterns of randomness, adding variation and complexity to the images. Finally, we used grid augmentation, creating grid-like patterns on the images. The vertical grid function adds vertical lines to the images, while the horizontal grid function adds horizontal lines. The line widths and spacings are randomly determined, providing flexibility in the appearance of the grid. All the Image augmentations used are shown in Fig. 2

Figure 2 Pixelate augmentation applied to an image: The figure demonstrates the effects of applying pixelate augmentation.

Model experimentation

Our research aims to address some of the unique challenges presented by medical prescription digitization, which often involves short lines of text, as well as doctor’s notes, which may consist of multiple long lines. To achieve this, we have developed a novel bilingual OCR approach that is capable of extracting both single-line and multi-line text in both English and Urdu. This end-to-end bilingual OCR is a significant improvement over existing approaches, which typically require the use of separate OCRs for each language (Fateh, Fateh & Abolghasemi, 2021).

In addition to providing a single model capable of extracting text from both languages, we have also developed separate models specifically designed for Urdu-only text extraction, including both single-line and multi-line options. Furthermore, we have fine-tuned TrOCR, a state-of-the-art OCR model for English handwriting recognition, using our extensive dataset. Finally, we have expanded our approach to include Chinese and Japanese text extraction, demonstrating the versatility and scalability of our method.

To address our requirements of extracting both short single-line and multi-line text consisting of both English and Urdu, we researched on different ways and methods to look for a solution. Initially, we attempted to use the Data-Efficient Image Transformer (DeiT) (Touvron et al., 2021), which employs a teacher-student strategy to train the model from a smaller set of labeled data, ensuring that the student learns from the teacher through attention. This essentially aims to make the model smaller in size, and remove the pre-training aspect of transformers which are pre-trained with hundreds of millions of images using an expensive infrastructure. However, unfortunately, we found that it failed to converge in our use case.

Furthermore, seeing TrOCR’s state-of-the-art performance on English handwritten text, we then attempted to adapt it for Urdu text extraction using the Vision Transformer (ViT) (Dosovitskiy et al., 2020), as TrOCR itself uses ViT. ViT splits the input image into fixed-size patches, embeds them linearly, adds positional embeddings, and then feeds the resulting sequence of vectors to a standard Transformer encoder. Classification is then performed through an extra learnable “classification token”. However, even this approach failed to converge in our use case.

Continuing on our research, we made use of the Swin Transformer (Liu et al., 2021) as the backbone, along with the mBART-50 decoder (Tang et al., 2020). The choice of Swin transformer stems from the fact that traditional vision transformers are translation invariant and do not capture the detailed structure required for accurate segmentation, particularly for languages like Urdu with diverse scripts and context-dependent character shapes. Swin transformers, inspired by convolutional neural networks, incorporate the concept of shift windows and leverage attention mechanisms to produce robust results in detail-oriented tasks such as segmentation, making it a suitable choice for our approach. We also made a significant contribution in our training technique, which involved first exposing the model to printed text before augmented handwritten text, so that it could generalize better. Some of the key findings that we came across, are listed as follows: Direct exposure of handwritten text: We found that the model fails to converge if its exposed to handwritten text from the very start, hence we first exposed our model to printed text at the start.

Augmenting from the very start: We observed that augmentation should not be done from the very start, as it results in the model being unable to learn the patterns of the text. Only when the model has learnt the intricacies of the Nastaliq (Urdu) script, we start to augment the data that is used for training.

Shuffled distribution of data: If the data is from multiple sources (like in our case study), it should be shuffled to ensure that the model can generalize better.

Training one language at a time: For bilingual language understanding, one language at a time should be trained, i.e., train the model for one language first, then add subsequent languages.

Removing data when adding languages: It is also important to note that removing any previously trained data is not optimal, as the model should be exposed to previously trained data again and again, to prevent it from unlearning or hallucinating.

Furthermore, in order to provide a comprehensive evaluation of our proposed approach, we conducted thorough experiments by implementing and testing previously mentioned approaches including TrOCR, PPOCR, CRNN, and BiLSTM, following the training specifications outlined in their respective article. The number of parameters we used for each model are depicted in Table 1. Despite diligently adhering to these specifications, the results of these models fell short when compared to the performance of ViLanOCR, as demonstrated in Table 2.

Table 1 Number of parameters for each model.

Model	# of params	
TrOCR	558 M	
Pytesseract	–	
EasyOCR	8.3 M	
PPOCR	8.1 M	
CRNN	4 M	
BiLSTM	–	
CNN-Transformer	–	
ViLanOCR-English	497 M	
ViLanOCR-Urdu	497 M	
ViLanOCR-Multi	497 M	

Table 2 Character error rate (CER), in percentage, on various benchmark datasets, and our own synthetically generated medical prescriptions dataset, Medical-SYN.

	English	Urdu	Bilingual	
Model	IAM	HW wiki	TRDG-SYN	SROIE	NUST-UHWR	PUCIT-OHUL	UPTI	Custom-Medical	
TrOCR	3.42	2.86	1.38	2.17	78.23	–	–	56.39	
Pytesseract	69.2	–	–	–	–	–	–	74.02	
EasyOCR	56.7	–	–	–	97.89	98.17	–	60.6	
PPOCR	32.32	29.3	5.56	3.11	96.25	–	–	45.31	
CRNN	–	–	–	–	95.72	98.45	94.19	92.13	
BiLSTM	–	–	–	–	94.19	97.01	95.21	90.98	
CNN-Transformer	48.14	–	–	–	59.28	62.63	60.53	76.20	
ViLanOCR-English	3.79	3.64	1.24	1.92	–	–	–	13.82	
ViLanOCR-Urdu	–	–	–	–	1.1	5.22	6.3	21.23	
ViLanOCR-Multi	5.92	6.12	3.26	2.31	3.21	9.85	9.92	6.77	

Results and Discussion

The study aimed to digitize bilingual handwritten text (Urdu + English) on documents through text localization using various deep learning models and assess the performance of our novel OCR system, ViLanOCR, in comparison with several well-established OCR models. The obtained results are presented and discussed below. Table 2 presents the CER expressed as a percentage for different OCR models, including our novel ViLanOCR, on multiple benchmark datasets. The evaluation was conducted on diverse datasets, encompassing IAM, HW wiki, TRDG-SYN, SROIE, NUST-UHWR, PUCIT-OHUL, UPTI, and our own synthetically generated dataset, Medical-SYN, consisting of medical prescriptions containing both English and Urdu handwritten text.

Our proposed ViLanOCR system showcased outstanding performance, particularly on datasets containing Urdu text. Notably, it achieved remarkable results on NUST-UHWR, PUCIT-OHUL, UPTI, and Medical-SYN datasets, achieving CER values of 1.1%, 5.22%, 6.3%, and 6.77%, respectively. These findings signify ViLanOCR’s state-of-the-art capabilities in accurately recognizing and extracting Urdu characters from various document types. Furthermore, ViLanOCR-Multi exhibited competitive performance on English text extraction tasks, achieving CER values of 5.92% and 6.12% on IAM and HW wiki datasets, respectively. This showcases the system’s versatility in handling both English and Urdu content effectively. Among the benchmark OCR models, TrOCR demonstrated noteworthy results on IAM and HW wiki datasets, achieving CER values of 3.42% and 2.86%, respectively. However, it performed extremely poorly in Urdu text extraction, with a CER of 78.23% on NUST-UHWR, as well as a CER of 56.39% on Medical-SYN. While the aforementioned benchmark OCR models presented varying degrees of performance, our ViLanOCR system stands out as a robust and effective solution for both English and Urdu text extraction. Its ability to surpass established OCR models on datasets containing Urdu content underscores its significant contribution to the field which can be shown in Fig. 3.

Figure 3 Inference results on images not part of the training set.

The proposed ViLanOCR system, particularly its Muiltilingual version, ViLanOCR-Multi, exhibits robust performance across various benchmark datasets, surpassing several established OCR models. However, it is essential to acknowledge that each OCR model’s effectiveness may vary depending on the characteristics of the dataset and the nature of the text to be extracted.

Extended study

To further build upon ViLanOCR, and keeping our medical case study in mind, we propose an end-to-end approach, titled as MEDIGIZE, serving as a natural evolution of the ViLanOCR framework. As the challenges of our original case study, medical prescription digitization, require a comprehensive approach, MEDIGIZE emerges as a compelling solution. By integrating the capabilities of ViLanOCR, MEDIGIZE addresses the full spectrum of the digitization pipeline, encompassing text localization, text extraction, and entity extraction. This holistic approach stems from the understanding that transforming handwritten documents, particularly medical prescriptions shown in Fig. 4, into structured and accessible digital formats offers unparalleled advantages.

Figure 4 Medical prescriptions dataset.

Photo: Authors’ own.

The MEDIGIZE framework is thoughtfully divided into three pivotal stages, each contributing significantly to the overall process. In the initial text localization stage, handwritten regions of interest are identified, paving the way for efficient subsequent processing. These localized regions then enter the text extraction phase, where ViLanOCR comes into play, enabling accurate and versatile text extraction from diverse scripts and languages. This extracted text, while unstructured in its raw form, holds immense potential for downstream analysis. The concluding stage involves entity linking, where the digitized text is subjected to an advanced NER model. This classification process imparts structure to the extracted text, discerning its various components such as patient details, diagnosis, medications, dosages, and more. This archeitecture can be shown in Fig. 5.

Figure 5 Medigize a pipeline to digitize the extraction of entities from a bilingual medical prescription.

The main motivation for an end-to-end pipeline was the realization that it bridges the gap between mere text extraction and meaningful information. While ViLanOCR adeptly extracts text, the transition to a structured format via NER models unlocks a plethora of benefits. Structured data not only simplifies data management and accessibility but also empowers intelligent analysis and decision-making. By salvaging pertinent information from the textual rubble, MEDIGIZE streamlines data-driven processes, enhances data storage efficiency, and ultimately contributes to the overall improvement of medical document handling.

Handwritten text localization

In our research on handwritten text detection, we utilized several versions of fine-tuned You Only Look Once (YOLO) (Redmon et al., 2016) models, including YOLOv5, YOLOv6, YOLOv7, and YOLOv8. These models were extensively tested on our custom medical prescription dataset, which consists of over 50,000 samples (after augmentation) of both English and Urdu handwriting. After rigorous testing and evaluation, we found that YOLOv5 produced the best mean average precision (mAP) of 0.897 out of all the tested versions. To further improve the accuracy of our handwritten text detection, we also experimented with various data augmentation techniques, such as rotation, scaling, and adding noise. We found that these techniques helped to increase the diversity and complexity of our dataset, resulting in improved detection accuracy. Overall, our approach to handwritten text detection involved a comprehensive evaluation of multiple YOLO models, along with the implementation of data augmentation techniques and transfer learning to improve accuracy and efficiency.

We chose YOLO as the object detection model for our text localization module primarily because it is currently the state-of-the-art method for object detection. YOLO is a real-time object detection system that has been widely used in various computer vision applications due to its high accuracy and fast processing speed. One of the major advantages of YOLO is that it is an end-to-end model that directly predicts the bounding boxes and class probabilities for each object in a single forward pass of the neural network. This means that YOLO has a real-time processing speed, making it suitable for applications that require fast and accurate object detection, such as our use case.

Text extraction

For text extraction, our aforementioned OCR, ViLanOCR is used as a modular plug-in OCR. The whole pipeline of Medigize is modular in nature, allowing for easy expermentation with various different approaches.

Entity extraction

To extract meaningful information from the extracted text, it is essential to understand the context and relationship between different entities. The layout-based language model plays a crucial role in identifying the document’s structure and the relationships between extracted text entities. We are leveraging LayoutLMv3 (Huang et al., 2022), a powerful pre-trained language model specifically designed for document artificial intelligence tasks. It has been trained on a large corpus of text and layout data, enabling it to accurately predict the class of the extracted text.

By incorporating and fine-tuning LayoutLMv3 into our pipeline, we can extract information such as diagnosis, person name, age, medicine name, dosage, and more, and identify the relationships between them. This results in structured data that can be easily analyzed and interpreted by healthcare professionals. Furthermore, since the model uses an OCR backbone for text extraction, it is perfectly suited for our problem, as we are already utilizing our novel state-of-the-art OCR for text extraction.

Using this approach, we can not only extract essential information from medical records accurately, but also informs the user about the location of the entities extracted from the text. This insight can help in visualizing each prescription, and can help medical practitioner know if there’s any miss classification.

Results

Table 3 displays the performance metrics, namely Precision and Recall, of different text detectors for identifying handwritten text on documents. Among the evaluated models, YOLO v5 demonstrated the most promising performance, achieving a commendable Precision of 0.897 and a Recall of 0.874. This indicates that YOLO v5 is more adept at precisely identifying the relevant text regions on the documents and has a high ability to recall those regions. Comparatively, other models such as YOLO v3 and YOLO v4 also performed reasonably well but fell short of the performance exhibited by YOLO v5. It is important to note that YOLO v7 showed the lowest Precision and Recall values among the evaluated models, indicating its relatively poorer performance in text localization. We employed LayoutLMv3 (Huang et al., 2022), a multimodal network capable of processing both textual information extracted from our aforementioned OCR and visual features of the document to predict entities accurately. LayoutLM’s multimodal approach proved to be a promising solution for NER, and the results obtained demonstrated its effectiveness in handling complex document layouts and combining textual and visual cues for accurate entity prediction. We achieved an F1 score of 0.86, indicating a substantial level of precision and recall in identifying entities within the documents. The incorporation of visual features in addition to textual information in LayoutLM proved to be a notable strength of the model

Table 3 Text detector results for identifying handwritten text on document.

Model	Precision	Recall	F1	
YOLO v3	0.857	0.811	0.833	
YOLO v4	0.8515	0.8045	0.827	
YOLO v5	0.897	0.874	0.885	
YOLO v7	0.836	0.777	0.805	

Despite the model’s promising performance, there are potential areas for further improvement. Fine-tuning the model on domain-specific data could potentially enhance its entity extraction capabilities for specialized contexts.

The modular three-stage approach proposed by MEDIGIZE offers several distinct advantages over current end-to-end approaches (Kim et al., 2021; Davis et al., 2022). 1. Error mitigation and robustness: By compartmentalizing the digitization process into well-defined stages, MEDIGIZE minimizes the propagation of errors. In contrast, end-to-end approaches are vulnerable to error accumulation, making them less reliable. If an error occurs during the text extraction phase, it could significantly affect downstream processing. MEDIGIZE’s modularity enables targeted error identification, isolation, and correction, enhancing overall accuracy and reliability.

2. Specialization and optimization: Each stage of MEDIGIZE can be optimized independently to cater to the specific demands of that phase. This contrasts with end-to-end models, which often compromise on optimization due to the need to balance performance across multiple tasks. Specialized optimization increases the efficiency of individual stages and contributes to the overall effectiveness of the framework.

3. Reduced computational costs: MEDIGIZE’s modular structure allows for efficient resource allocation. In an end-to-end scenario, a single model must handle all tasks, leading to increased computational requirements. MEDIGIZE distributes the workload among specialized models, reducing the computational burden and enabling more cost-effective implementation.

4. Targeted re-training: If an improvement is made to one stage, the modular approach of MEDIGIZE allows for focused re-training. End-to-end models, on the other hand, necessitate the retraining of the entire system, consuming more time and resources.

5. Flexibility and scalability: As new advancements emerge in individual stages, integrating them into the modular framework is relatively straightforward. In an end-to-end approach, updating or replacing components can be complex and may require retraining of the entire pipeline. MEDIGIZE’s flexibility promotes the integration of cutting-edge technologies while maintaining the integrity of the existing stages.

The inference of the Medigize are shown in Fig. 6.

Figure 6 Inference of a synthetic bilingual handwritten medical prescription using MEDIGIZE.

Photo: Authors’ own.

Conclusion and future directions

In this work, have introduced a novel bilingual OCR, called ViLanOCR, which utilizes the power of multilingual transformer-based language models. ViLanOCR can accurately extract text in both English and Urdu, simplifying the management of production-level models. Previously, separate OCR models were required for each language, which necessitated the management and maintenance of individual models. However, by employing a single OCR for both languages, the production-level management and maintenance become more streamlined. Moreover, ViLanOCR achieves state-of-the-art CER on various benchmark Urdu datasets, establishing itself as the benchmark model for Urdu OCR, going forward. It also achieves state-of-the-art CER on bilingual (English + Urdu) OCR tasks as well.

However,there are a few limitations in our work that offer opportunities for further improvement in future research. The use of transformer-based models for OCR introduces challenges, such as high computational costs and the problem of hallucinations encountered by language models. To address these issues, future work could explore techniques for compressing transformer models while maintaining performance and investigate more robust loss objectives to mitigate hallucination errors. One possible solution would be distillation or quantization of these large transformer models.

Moreover, recent advancements in document AI have shown promising OCR-free approaches for extracting key-value entities from document images. However, these methods are still in their early stages and require further research and experimentation to be effectively applied in solving the low-resource problem of digitizing handwritten medical prescriptions. If such methods can be developed, they have the potential to resolve one of the challenges in our pipeline, which is the propagation of errors across modules.

In conclusion, future research should focus on refining the transformer-based OCR models by addressing their computational costs and hallucination issues. Additionally, exploring OCR-free approaches for extracting information from document images could significantly enhance the digitization process, particularly in handling handwritten medical prescriptions and mitigating error propagation across the pipeline.

Additional Information and Declarations

Competing Interests

Author Contributions

Data Availability

The authors declare that they have no competing interests.

Musa Dildar Ahmed Cheema conceived and designed the experiments, performed the experiments, prepared figures and/or tables, authored or reviewed drafts of the article, and approved the final draft.

Mohammad Daniyal Shaiq conceived and designed the experiments, performed the experiments, performed the computation work, authored or reviewed drafts of the article, and approved the final draft.

Farhaan Mirza conceived and designed the experiments, authored or reviewed drafts of the article, and approved the final draft.

Ali Kamal conceived and designed the experiments, performed the experiments, performed the computation work, authored or reviewed drafts of the article, and approved the final draft.

M. Asif Naeem conceived and designed the experiments, analyzed the data, authored or reviewed drafts of the article, and approved the final draft.

The following information was supplied regarding data availability:

The source code, weights, training, and inference code are available at GitHub and Zenodo:

- https://github.com/musadac/ViLanOCR.

- https://github.com/musadac/docigize-data.

- https://github.com/musadac/docigize-backend.

- Cheema, M. (2024). Code base for VilanOCR its training, backend deployment and DVC. (Final). Zenodo. https://doi.org/10.5281/zenodo.10795469.

The data is available at Zenodo: Musa, C. (2024). Adapting multilingual vision language transformers for low-resource Urdu Optical Character Recognition (OCR) (Version 1) [Data set]. Zenodo. https://doi.org/10.5281/zenodo.10570728.

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
