# Peer review of "Adapting multilingual vision language transformers for low-resource Urdu optical character recognition (OCR)"

_PeerJ Computer Science, doi:10.7717/peerj-cs.1964_

## Round 0.1 · original submission · Major Revisions

Reviewers found merit in the paper and suggested a major revision. You are required to revise your manuscript and address all the comments and suggestions of reviewers. I also have a few minor suggestions:

1) In the abstract, it mentioned that "This paper addresses the significance of focusing on such languages and introduces ViLanOCR, an innovative bilingual OCR system tailored for Urdu and English". Further, it is mentioned that "ViLanOCR leverages advanced multilingual ...". Hence, bilingual and multilingual contradicts. Authors must resolve it. Further, remove the sentence "We provide the source code, weights, training, and inferences of our models available to the reviewers." from the abstract.

2) Referencing in the entire manuscript has not been done properly. Authors are required to follow journal guidelines for the same.

3) Section 3: "PROPOSED APPROACH" has been written very concisely which is hard to follow. Authors are required to extend this section significantly, specially about the working of ViLanOCR (Vision Language-based OCR).

4) What is the difference between Section 5 (Results & Discussion) and Section 6 (Extended Study)? I would suggest to merge these two sections under "Results & Discussion).

5) Most of the Figures are not cited within the text.

**Language Note:** PeerJ staff have identified that the English language needs to be improved. When you prepare your next revision, please either (i) have a colleague who is proficient in English and familiar with the subject matter review your manuscript, or (ii) contact a professional editing service to review your manuscript. PeerJ can provide language editing services - you can contact us at copyediting@peerj.com for pricing (be sure to provide your manuscript number and title). – PeerJ Staff

Reviewer 1 ·

Basic reporting

Authors have proposed a system namely ViLanOCR, a bilingual OCR system for Urdu and English.
The literature review must be enhanced with some more latest references.
There are some typographical and grammatical mistakes. They need to be corrected.
Use high resolution images only.
The equations should be typed only in equation editor if any.

Experimental design

The main contribution of the paper must be mentioned in bulleted form in the introduction section
The work done by Anjum and Khan (2020) may be written in compact form..similarly for Ul et al. ul Sehr Zia et al. (2022). There is too much of details which readers are not interested to go through. Compact them in a form which convey the proper information.
For line 231 "To validate the efficacy of ViLanOCR, we focus on the domain of handwritten medical prescriptions", may include the snapshot of the data set in the manuscript i.e., handwritten medical prescriptions
The nearest-neighbor interpolation technique used shall have upscaled images but with jaggies like edges. How was that addressed?
How where features ranked

Validity of the findings

Metrics used for evaluation need to be highlighted
What are the limitations of the study. Mention that at the end.

Additional comments

No comments

Reviewer 2 ·

Basic reporting

Its a well-written article with good scientific content and professional and scientific structure.

Experimental design

The experimental design is appropriate.

It is recommended is show the images associated with different augmentations employed

recommended to evaluate F1 score for yolo variants

Validity of the findings

Seems valid

Additional comments

The manuscript is well written.
It is recommended to include a comparison of computational costs

---

## Round 0.2 · Minor Revisions

Dear Authors,
Reviewer 2 has raised the following concerns:
"I could not find the response to my comments, document. There is a rebuttal document (cs-91551-Rebuttal) but it does not list my comments although these were minor. The authors have included some of my comments in the revision whereas some are not incorporated although these were minor they could improve the quality of the manuscript. IF i am missing some files kindly forward them to me."

You are required to address these concerns and resubmit it.

Reviewer 1 ·

Basic reporting

All the comments have been addressed and the manuscript seems in good shape now.

Experimental design

no comment

Validity of the findings

no comment

Reviewer 2 ·

Basic reporting

ok

Experimental design

ok

Validity of the findings

ok

Additional comments

I could not find the response to my comments, document.
There is a rebuttal document (cs-91551-Rebuttal) but it does not list my comments although these were minor.

The authors have included some of my comments in the revision whereas some are not incorporated although these were minor they could improve the quality of the manuscript.

IF i am missing some files kindly forward them to me.

---

## Round 0.3 · accepted · Accept

The authors have revised the manuscript as per the suggestions of reviewers and academic editor. Now, I am happy to accept this paper for publication in its current form.